# Sex steroids regulate skin pigmentation through nonclassical membrane-bound receptors

Christopher A Natale[1], Elizabeth K Duperret[1], Junqian Zhang[1], Rochelle Sadeghi[1], Ankit Dahal[1], Kevin Tyler O'Brien[2], Rosa Cookson[2], Jeffrey D Winkler[2], Todd W Ridky[1]\*

[1]Department of Dermatology, Perelman School of Medicine, University of Pennsylvania, Philadelphia, United States; [2]Department of Chemistry, University of Pennsylvania, Philadelphia, United States

**Abstract** The association between pregnancy and altered cutaneous pigmentation has been documented for over two millennia, suggesting that sex hormones play a role in regulating epidermal melanocyte (MC) homeostasis. Here we show that physiologic estrogen (17β-estradiol) and progesterone reciprocally regulate melanin synthesis. This is intriguing given that we also show that normal primary human MCs lack classical estrogen or progesterone receptors (ER or PR). Utilizing both genetic and pharmacologic approaches, we establish that sex steroid effects on human pigment synthesis are mediated by the membrane-bound, steroid hormone receptors G protein-coupled estrogen receptor (GPER), and progestin and adipoQ receptor 7 (PAQR7). Activity of these receptors was activated or inhibited by synthetic estrogen or progesterone analogs that do not bind to ER or PR. As safe and effective treatment options for skin pigmentation disorders are limited, these specific GPER and PAQR7 ligands may represent a novel class of therapeutics.

\*For correspondence: ridky@mail.med.upenn.edu

## Introduction

Cutaneous pigmentary changes have been long recognized as common side effects of pregnancy. The British physician Daniel Turner, in his 1714 *De Morbis Cutaneis* (*Turner, 1726*), references Hippocrates (460–370 B.C.E.), "There is a spot on the face…more peculiar, according to our great master *Hippoc.,* to Big Belly'd women, and recon'd as one of the Signs of Conception." Modern physicians recognize this common pregnancy-associated hyperpigmentation as melasma (*Sheth and Pandya, 2011*; *Nicolaidou and Katsambas, 2014*). Hippocrates also thought that the pigment was predictive of the sex of the fetus: *Quae utero gerentes, maculum in facie veluti ex solis adustione habent, eae faemellas plerumque gestant.* Translated to English: *pregnant women who have a mark on the face as though stained by the sun, quite often give birth to girls.* While Turner noted this association with fetal sex to be 'fallible', Hippocrates was remarkably astute in linking the pigment increases to the tanning response to DNA-damaging solar ultraviolet (UV) radiation. While early physicians attributed the pigment changes to "Retention of the menstrual Flux" (*Turner and Cutaneis, 1726*), the molecular mechanisms through which pregnancy-associated hormonal changes modulate skin color have remained elusive for over 2,000 years.

Melanocytes in the basal epidermis control skin pigmentation through synthesis of melanin, a complex process thought to be primarily regulated by alpha-melanocyte stimulating hormone (αMSH) (*Figure 1—figure supplement 1A and B*). The αMSH peptide is secreted centrally by the anterior pituitary gland, and locally by keratinocytes in response to UV damage (*Cui et al., 2007*). αMSH binding to the melanocortin receptor 1 (MC1R), a G protein-coupled receptor (GPCR),

**eLife digest** Factors controlling pigment production in skin are complex and poorly understood. Cells called melanocytes produce a pigment called melanin, which makes the skin darker. It has been known for a long time that skin color often changes during pregnancy, which suggests that sex hormones may be involved. However, the specific hormones and signaling mechanisms responsible for the changes have remained largely undefined.

Estrogen and progesterone are two of the main female sex hormones. Natale et al. now show that estrogen increases pigment production in human melanocytes, and progesterone decreases it.

For hormones to signal to cells, they must bind to and activate particular receptor proteins. Further investigation by Natale et al. revealed that estrogen and progesterone regulate pigment production by binding to receptors that belong to a family called G protein-coupled receptors. These receptors can signal rapidly once activated by sex hormones, and may serve as therapeutic targets for treating pigmentation disorders.

Skin diseases that cause inflammation often also cause changes in skin color. Natale et al. noticed several other G protein-coupled receptors that are likely to control pigmentation through similar mechanisms. Future analyses of the roles that these other receptors perform in melanocytes may therefore reveal how inflammation-based pigmentation changes occur.

activates adenylate cyclase, and increases cAMP. This secondary messenger activates a cascade of downstream transcriptional events leading to expression of genes required for melanin synthesis (*Rodríguez and Setaluri, 2014*). Exogenous broadly-acting adenylate cyclase activators such as plant-derived forskolin, also stimulate melanin production (*D'Orazio et al., 2006*), but the degree to which other endogenous molecules, other than αMSH regulate melanin synthesis in tissue is unclear. However, the observation that melasma frequently occurs in non-pregnant women using oral contraceptive pills, which contain only steroid hormone analogs (*Sheth and Pandya, 2011*; *Resnik, 1967a*), suggests that humans may maintain αMSH-independent pigment control mechanisms. Identifying these pathways, and strategies to specifically access them pharmacologically to modulate skin pigmentation, may have substantial therapeutic utility.

## Results

### Estrogen and progesterone reciprocally regulate melanin synthesis

To examine whether sex steroids influence melanin synthesis, we treated primary human melanocytes with estrogen (17β-estradiol). This resulted in a dose-dependent melanin increase (*Figure 1—figure supplement 1C*). After 4 days of exposure to 25 nM estrogen, a medically-relevant concentration observed during pregnancy (*Abbassi-Ghanavati et al., 2009*), melanin was markedly increased (208% +/- 27%) in three individual isolates of primary human melanocytes (*Figure 1A*). The magnitude of this change was similar to that observed with αMSH (*Figure 1—figure supplement 1D*), and is consistent with prior in vitro studies implicating estrogen in melanin synthesis (*McLeod et al., 1994*; *Ranson et al., 1988*). Hormonal oral contraceptives, most of which incorporate ethinyl estradiol, are associated with melasma (*Resnik, 1967b*). Ethinyl estradiol also increased melanin to levels similar to those observed with native estrogen. To examine the effects of estrogen on melanocyte homeostasis in the context of intact human epidermis, architecturally-faithful three-dimensional organotypic skin was established utilizing normal primary epidermal keratinocytes and melanocytes in native human stroma (*Ridky et al., 2010*; *Monteleon et al., 2015*; *Duperret et al., 2014*; *McNeal et al., 2015*). After one week, estrogen-treated skin displayed a threefold increase in melanin content (*Figure 1B*), without changes in melanocyte number or density (*Figure 1C*).

Estrogen effects in other tissue types are often counter-balanced by progesterone (*Ismail et al., 2015*), which also increases during pregnancy. To determine whether this reciprocal relationship is active in melanocytes, we treated cells with physiologic levels of progesterone, which resulted in a dose-dependent decrease in melanin production (*Figure 1—figure supplement 1F*). At 500 nM, a concentration observed in third trimester pregnancy (*Abbassi-Ghanavati et al., 2009*), progesterone

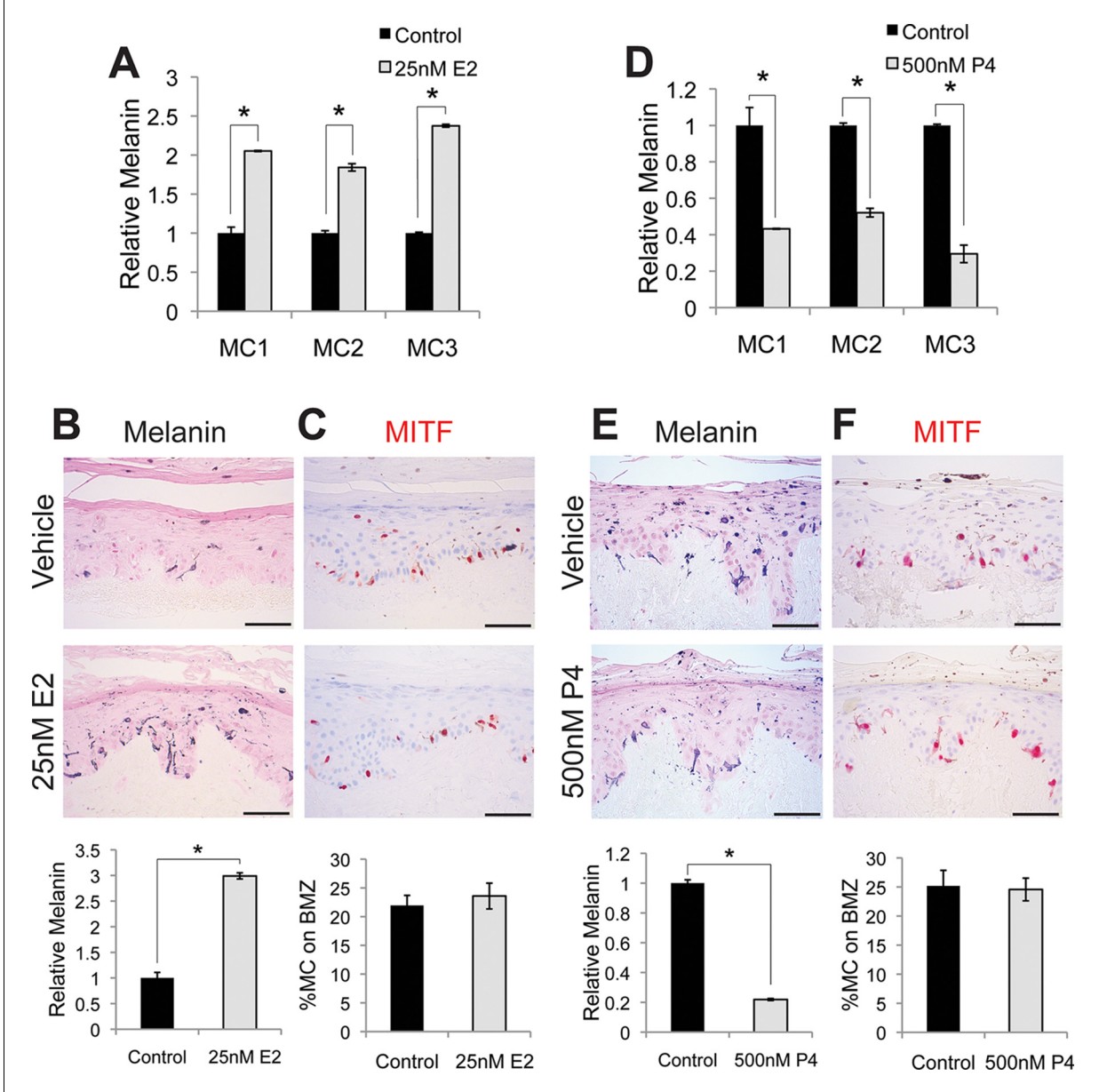

**Figure 1.** Estrogen and progesterone reciprocally regulate melanin synthesis. (A) Melanin content of primary human melanocytes treated with estrogen (E2), compared to vehicle-treated controls. (B) Fontana-Masson (melanin) staining of organotypic skin treated with vehicle or estrogen. Relative melanin content is quantified below. (C) MITF immunohistochemistry of organotypic skin treated with vehicle or estrogen. Melanocyte population density is quantified below. (D) Melanin content of primary human melanocytes treated with progesterone (P4), compared to vehicle. (E), Fontana-Masson (melanin) staining of organotypic skin tissues treated with progesterone or vehicle. Relative melanin content is quantified below. (F) MITF immunohistochemistry of organotypic skin tissues treated with vehicle or progesterone. Melanocyte population density is quantified below. n=3 biologic replicates for each experiment. Error bars denote +/- s.d., *p<0.05, scale bar = 50 µm.

The following figure supplements are available for figure 1:

**Figure supplement 1.** Melanin production in melanocytes.

**Figure supplement 2.** Relative proliferative response to estrogen and progesterone treatment.

decreased melanin production by half (58% +/- 11.4%), both in culture (*Figure 1D*) and in skin tissue (*Figure 1E*), without altering melanocyte cell number (*Figure 1F*). Most of our primary melanocytes were derived from newborn male foreskin. To determine whether female cells also responded similarly, we treated female iPS-derived melanocytes with estrogen and progesterone and noted responses similar to those observed with the male cells (*Figure 1—figure supplement 1G*). To determine whether melanocytes isolated from body sites other than foreskin also responded similarly to sex hormones, we treated melanocytes from adult facial skin with estrogen and progesterone and again observed responses that were similar to those observed with the foreskin melanocytes (*Figure 1—figure supplement 1H*).

Consistent with other groups who have noted that steroid hormones have variable effects on melanocyte proliferation in culture (*Im et al., 2002*), we observed modest changes in proliferation when isolated primary cells were treated with estrogen or progesterone in vitro. Estrogen treated melanocytes tended to proliferate slightly slower, while progesterone treated cells tended to proliferate slightly faster (*Figure 1—figure supplement 2A–B*). The effects varied with the basal level of melanin production. Melanocytes from dark skin were more sensitive to progesterone than estrogen, while melanocytes from light skin were more sensitive to estrogen. These proliferation changes are likely an in vitro artifact, as adult interfollicular epidermal melanocytes are relatively nonproliferative in vivo, and we did not note any changes in melanocyte numbers in sex steroid-treated 3-D organotypic tissues. Consistent with this lack of melanocyte proliferation in interfollicular epidermis, another group thoroughly examined 280 tissue sections from normal human skin from 18 donors, and identified only 2 proliferative interfollicular melanocytes (*Jimbow et al., 1975*).

## Primary human melanocytes do not express nuclear estrogen or progesterone receptors (ER/PR), and respond to sex steroids via altered cAMP signaling

To determine the mechanisms mediating estrogen and progesterone pigment effects, we examined components of the canonical pigment production pathway, and observed a cAMP increase upon estrogen treatment (*Figure 2A*), suggesting that estrogen accesses the canonical pigment production pathway downstream of MC1R. Consistent with this, pCREB and MITF proteins were similarly induced (*Figure 2B*). In contrast, progesterone reciprocally decreased melanin, cAMP, pCREB and MITF (*Figures 2C–D*).

These data indicate that estrogen, progesterone, and αMSH converge on adenylate cyclase to reciprocally modulate melanin synthesis, and suggest that individual steroid effects may counter-balance each other. Consistent with this, the estrogen effects were significantly attenuated in the presence of progesterone (*Figure 2E*). This likely helps explain why pregnancy-associated hyperpigmentation is characteristically limited to specific areas where melanocyte or UV radiation exposure is highest including the face, genital, and areolar regions (*Szabo, 1954*; *Staricco and Pinkus, 1957*). It is also possible that in the complex hormonal milieu of pregnancy, additional factors beyond the sex steroid activated pathways described here also contribute to skin color modulation.

As steroid hormones are not predicted to signal through MC1R, whose natural ligand is the peptide αMSH, we sought to identify the specific receptors mediating the estrogen and progesterone pigment effects. We did not detect classical estrogen (ER) or progesterone (PR) receptors in melanocytes using qRT-PCR, although transcripts were observed in breast cells (*Figure 2—figure supplement 1A*). Previous RNAseq studies in human melanocytes, conducted for unrelated experimental questions (*Flockhart et al., 2012*), also failed to detect ER or PR transcripts (*Figure 2—figure supplement 1B*). Consistent with this, ER or PR protein was not observed in MC via western blotting, although both receptors were readily apparent in breast cells (*Figure 2F*).

## Sex steroid signaling in melanocytes is dependent on the nonclassical membrane bound G-protein coupled sex hormone receptors GPER and PAQR7

Considering MC1R is a G protein-coupled receptor (GPCR), we hypothesized that alternative GPCRs mediate sex steroid pigment effects. To identify possible candidates, we analyzed whole transcriptome melanocyte RNAseq data. Of 412 known or predicted 7-pass human GPCRs (*Alexander et al., 2013*), 61 distinct GPCRs were expressed in MCs, including the membrane-

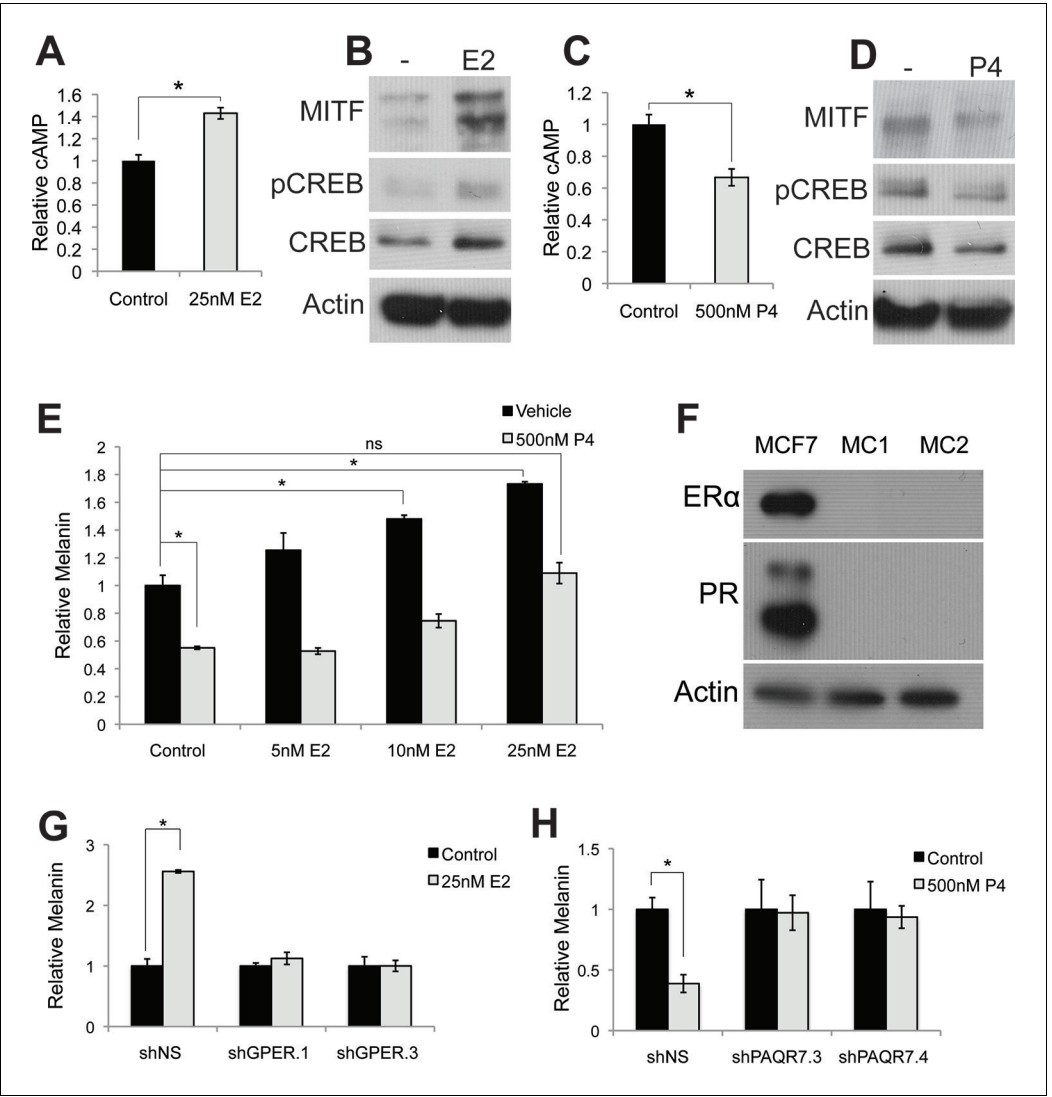

**Figure 2.** Estrogen and progesterone access the classical melanin production pathway through nonclassical receptors. (**A**) cAMP ELISA from estrogen-treated melanocytes (**B**) Western blot demonstrating changes in classical melanin pathway regulators after a 16 hr estrogen treatment. (**C**) cAMP ELISA from progesterone-treated melanocytes. (**D**) Western blot demonstrating changes in classical melanin pathway regulators after a 16 hr progesterone treatment. (**E**) Melanin assay from melanocytes treated with estrogen and progesterone simultaneously. (**F**) Western blot for estrogen and progesterone receptors in MCF7 cells and melanocytes. (**G**) Melanin content of melanocytes transduced with control shRNA or shRNA targeting GPER. Cells were treated with either vehicle or estrogen. (**H**) Melanin assay performed on melanocytes transduced with control shRNA or shRNA targeting PAQR7. Cells were treated with either vehicle or progesterone. n=3 biologic replicates for each experiment. Error bars denote +/- s.d., *p<0.05.

The following source data and figure supplements are available for figure 2:

**Source data 1.** List of GPCR transcripts expressed in primary human melanocytes.

**Figure supplement 1.** Hormone receptors in melanocytes.

**Figure supplement 2.** Progesterone signals through Gi in melanocytes.

bound, G protein-coupled estrogen receptor (*Filardo et al., 2002*) (GPER) (*Figure 2—source data 1* and *Figure 2—figure supplement 1B*). Given that prior work in breast cancer cell lines and fish oocytes determined that estrogen binding to GPER modulates cAMP (*Filardo et al., 2002*; *Thomas et al., 2005*; *Cabas et al., 2013*; *Majumder et al., 2015*; *Pang and Thomas, 2010*), and that cAMP signaling stimulates melanin synthesis, we thought it possible that GPER may be the physiologically relevant human melanocyte estrogen receptor. The melanocyte RNA-seq studies also demonstrated that an analogous, noncanonical G protein-coupled progesterone receptor, progestin and adipoQ receptor 7 (PAQR7) (*Zhu et al., 2003*; *Tang et al., 2005*), was also expressed (*Figure 2—figure supplement 1B*). We next used qRT-PCR to verify that GPER and PAQR7 are both expressed in primary human MC (*Figure 2—figure supplement 1C*). Notably, GPER and PAQR7 expression was markedly lower in other skin cells including keratinocytes and fibroblasts (*Figure 2—figure supplement 1D*).

To establish the necessity of GPER and PAQR7 in mediating sex hormone effects in MCs, we first depleted GPER using either of two independent shRNA hairpins, which completely eliminated the melanocyte pigmentation response to estrogen (*Figure 2G* and *Figure 2—figure supplement 1E–F*). Analogous shRNA-mediated PAQR7 depletion ablated the pigmentary response to progesterone (*Figure 2H* and *Figure 2—figure supplement 1F*). To verify these results, we next used a complementary genetic approach based on CRISPR-Cas9 mediated gene disruption of GPER or PAQR7, which also completely blocked the pigmentary response to estrogen and progesterone, respectively (*Figure 2—figure supplement 1G–H*). Consistent with our model in which these receptors access melanin synthesis at the level of adenylate cyclase, PAQR7 was found to bind progesterone and regulate the final stages of sea trout oocyte meiosis through cAMP reduction (*Zhu et al., 2003*; *Tokumoto et al., 2012*). In that fish system, PAQR7 signals through G protein complexes containing the inhibitory G subunit ($G_i$), which represses adenylate cyclase. To examine whether this signaling mechanism is functional in melanocytes, we treated melanocytes with progesterone in the presence of pertussis toxin (PTX), an exotoxin that specifically inactivates $G_i$ subunits via ADP-ribosylation. With PTX treatment alone, we observed a small increase in melanin production. This likely reflects inhibition of $G_i$ released from basal PAQR7 activity, as well $G_i$ subunits from other GPCRs that collectively contribute to the basal level of cAMP signaling observed in culture. Importantly, PTX blocked progesterone effects, establishing that progesterone signals through $G_i$ subunits (*Figure 2—figure supplement 2A*).

## Specific activation of GPER or PAQR7 is sufficient to alter pigment production in human skin tissue

To complement these genetic studies establishing that GPER and PAQR7 are the melanocyte sex steroid receptors, we utilized a pharmacologic approach employing synthetic steroid analogs with specific agonist or antagonist activity on ER, PR, GPER, or PAQR7. Tamoxifen, an ER antagonist, is associated with melasma in breast cancer patients (*Kim and Yoon, 2009*). The mechanistic basis for the pigment change was previously unknown. However, tamoxifen is a GPER agonist (*Thomas et al., 2005*; *Li et al., 2010*), and increased melanin to levels comparable to those observed with estrogen (*Figure 3—figure supplement 1A*). To determine whether GPER signaling was sufficient to increase melanin, we utilized the specific GPER agonist G-1 (*Bologa et al., 2006*), an estrogen analog developed for mechanistic studies in other systems that does not bind ER. G-1 drove a dose-dependent increase in melanin production through pCREB and MITF that was GPER dependent (*Figure 3A–C* and *Figure 3—figure supplement 2A–D*). Further establishing that GPER is the melanocyte estrogen receptor, G-1 and estrogen effects were blocked by either of two specific GPER antagonists, G-15 and G-36 (*Figure 3—figure supplement 2E*), which do not have inhibitory activity against ER (*Dennis et al., 2009*; *2011*). To establish that PAQR7 signaling is sufficient to decrease melanin production, we used a specific PAQR7 agonist Org OD-02 (CH2P4), which does not bind PR (*Kelder et al., 2010*). CH2P4 caused a dose-dependent decrease in melanin production through pCREB and MITF that was PAQR7 dependent (*Figure 3D–F* and *Figure 3—figure supplement 3A–D*).

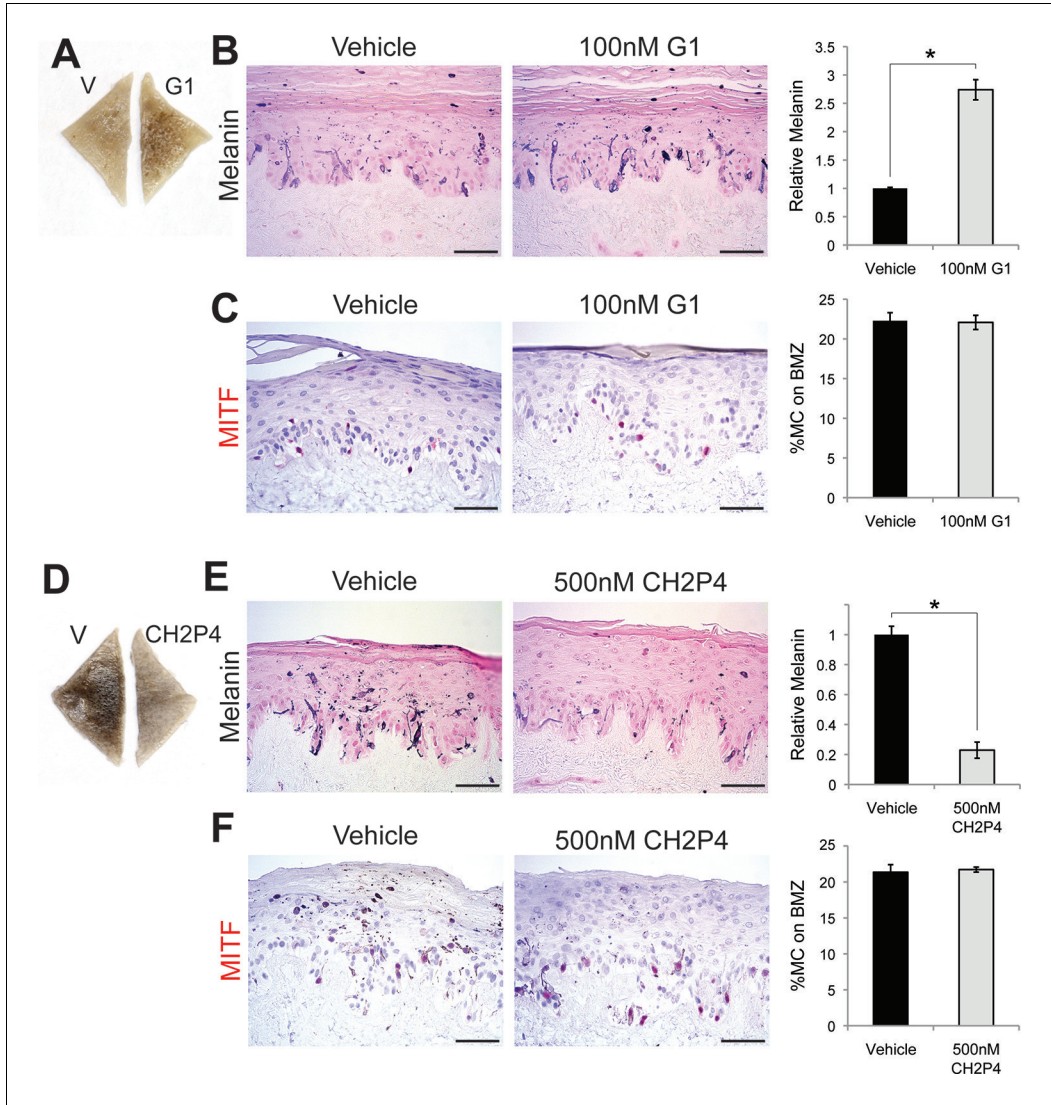

**Figure 3.** GPER and PAQR7 signaling is sufficient to alter melanin production in organotypic human tissue. (**A**) Organotypic skin treated with vehicle (left) or G-1 (right). (**B**) Fontana-Masson (melanin) staining of organotypic skin treated with vehicle or G-1. Quantification of melanin content is shown on the right. (**C**) MITF immunohistochemistry of organotypic skin treated with vehicle or G-1. Quantification of melanocyte population density is shown on the right. (**D**) Organotypic skin treated with vehicle (left) or CH2P4 (right). (**E**) Fontana-Masson (melanin) staining of organotypic skin treated with vehicle or CH2P4. Quantification of melanin content is shown on the right. (**F**) MITF immunohistochemistry of organotypic skin treated with vehicle or CH2P4. Quantification of melanocyte population density is shown on the right. n=3 biologic replicates for each experiment. Error bars denote +/- s.d., *p<0.05, scale bar = 50 μm.

The following figure supplements are available for figure 3:

**Figure supplement 1.** Melanin production is altered by sex steroid analogs currently in clinical use.

**Figure supplement 2.** Targeting GPER with specific agonists and antagonists.

**Figure supplement 3.** Targeting PAQR7 with specific agonists.

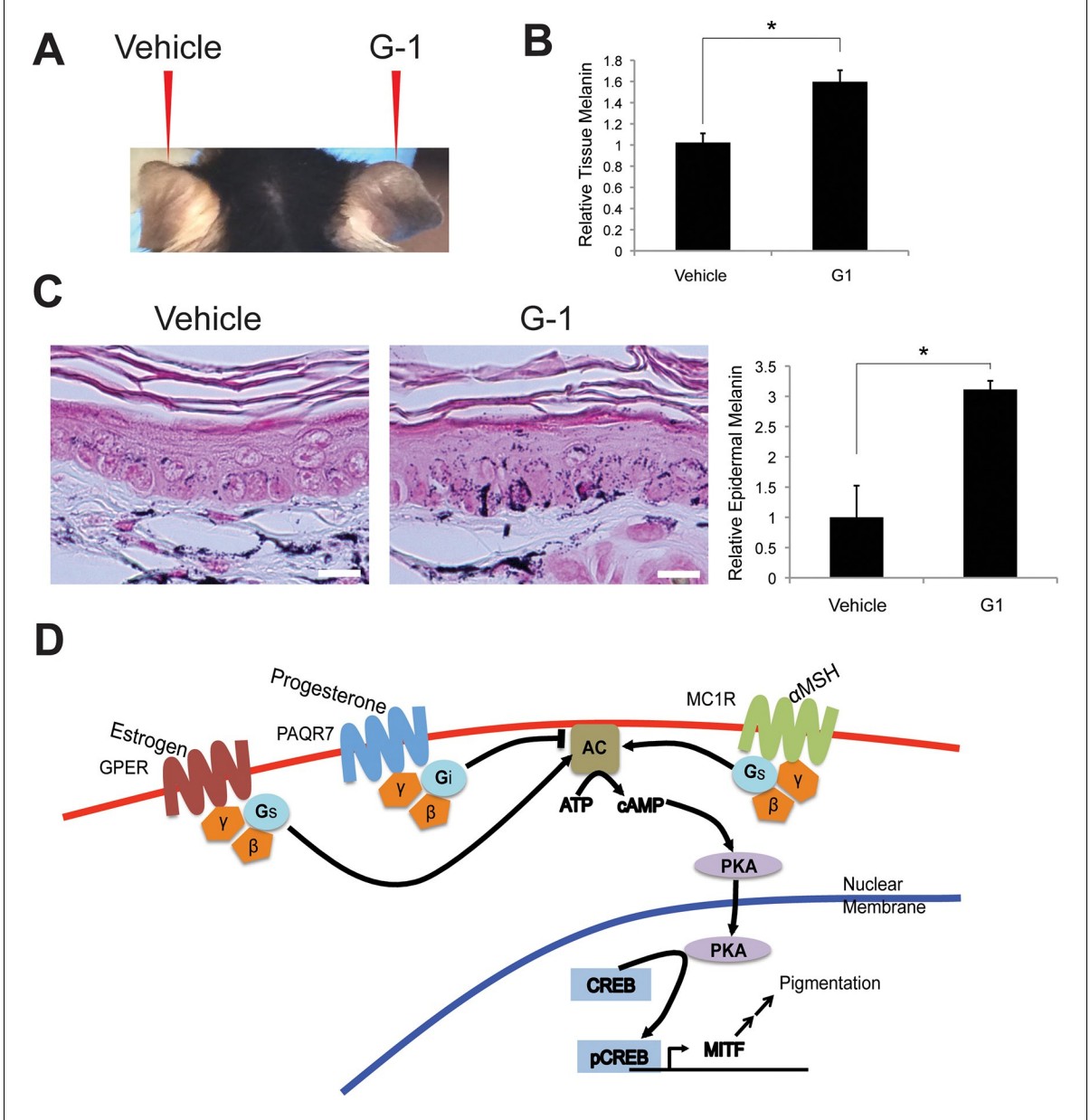

**Figure 4.** Topical GPER agonists increase pigmentation in vivo. (**A**) Mouse ear skin treated for 3 weeks with vehicle only on the left ear, and 2% (w/v) G-1 on the right ear. (**B**) Melanin assay on whole ear tissue that was treated with either vehicle or 2% G-1 for 3 weeks. (**C**) Fontana-Masson (melanin) staining of tissue sections from ears treated with either vehicle or 2% G-1, quantification of staining on right. (**D**) Schematic model of estrogen and progesterone signaling in melanocytes. n=3 biologic replicates for each experiment. Error bars denote +/- s.d., *p<0.05, scale bar = 20 μm.

The following figure supplement is available for figure 4:

**Figure supplement 1.** NMR spectrometry of synthesized G-1.

## Topical delivery of a GPER-specific synthetic estrogen analog in vivo increases epidermal melanin

To demonstrate that GPER was sufficient to promote melanin production in vivo, we synthesized G-1 to 95% purity (*Figure 4—figure supplement 1A–B*), and formulated G-1 for topical application. We treated the right ears of mice daily for 3 weeks with vehicle or 2% (w/v) G-1 in DMSO, and

observed a gradual increase pigmentation compared to vehicle-treated controls over 3 weeks (*Figure 4A*). Melanin content was increased 1.6-fold, a cosmetically significant change, and was consistent with the magnitude of change seen in vitro (*Figure 4B–C*). Clinically apparent skin darkening on mice increased over 2–3 weeks while pigment changes in culture were more rapid. It is likely that the synthetic GPCR ligands are metabolized and/or distributed differently in vivo then in in vitro culture, such that the effective local concentration of steroid in the in vivo setting is relatively transient. Optimization of an ideal topical formulation and dosing schedule will require additional study in the context of a human clinical trial.

## Discussion

Safe and effective approaches for modulating skin melanocyte function for therapeutic benefit are lacking, largely because the factors normally regulating melanocyte homeostasis are complex and incompletely deciphered. Defining these mechanisms is important however, as myriad genetic and acquired conditions including common afflictions such as acne, eczema, vitiligo, ultraviolet (UV) radiation exposure, traumatic injury, and pregnancy are associated with alterations in skin pigmentation that can be extensive and long-lasting (*James et al., 2011*). Another population that could potentially benefit from modulating skin pigment are people with naturally light skin, especially those with red hair, who have a markedly decreased ability to synthesize UV-protective brown eumelanin as a result of inactivating mutations in MC1R (*Valverde et al., 1995*). This large population is especially susceptible to photodamage, sunburns, and has an increased lifetime risk of keratinocyte and melanocyte-derived skin cancers (*Han et al., 2006*). There is currently no available therapeutic that promotes protective eumelanin pigment production. However, the specific activation of GPER alternatively activates cAMP signaling, bypassing MC1R, to stimulate melanin synthesis, and could therefore be especially useful in this sun-vulnerable population. Selective GPER activation in skin could potentially be a safe alternative to intentional UV radiation exposure (via natural sunlight or tanning beds) for individuals seeking what they perceive as an aesthetically desirable tan. The only method currently available to increase skin melanin is UV exposure. While effective at darkening skin, the requisite DNA damage promotes premature aging, wrinkles, and skin cancer.

Commonly utilized approaches for decreasing skin melanin are also often unsafe, and involve application of toxic mercury or arsenic compounds, especially common in India, China, Japan, and Korea, but also encountered in the U.S., and recently highlighted in a report from the California Department of Public Health (Report #14–046, 2014), or hydroquinone, a tyrosinase inhibitor, which has been banned in Europe because of concerns regarding its possible association with cancer (*McGregor, 2007*). Our findings describe small molecule sex steroid analogs, without these toxicities, that modulate pigment production (*Figure 4D*).

Therapeutic use of GPER or PAQR7 agonists/antagonists could potentially have effects on cells other than epidermal melanocytes. While topical delivery of such agents would likely avoid off target effects in distant tissues, there exits the theoretical possibility of off-target effects within the skin. However, we did not note any significant abnormalities in the epidermis from our in vitro or in vivo skin tissues treated with the sex steroids.

GPER and PAQR7 have been identified only relatively recently, but are expressed in several tissues, and may mediate at least some of the estrogen and progesterone effects that cannot be attributed to the classical nuclear hormone receptors. GPER has been identified in the reproductive, nervous, cardiovascular, renal, pancreas, and immune systems (*Prossnitz and Barton, 2011*). In immune cells, GPER expression on T cells has been shown to play a role in 17β-estradiol-induced thymic atrophy and autoimmune encephalomyelitis. PAQR7 is expressed in the reproductive and nervous systems (*Tokumoto et al., 2016*), and in murine macrophages (*Lu et al., 2015*) and in human T cells (*Dosiou et al., 2008*), although the functional role of PAQR7 in those tissues remains relatively unclear. Given that the increased systemic estrogen and progesterone associated with pregnancy does not typically result in skin cancer or significant pathology in other tissues, we think it likely that the specific GPER and PAQR7 agonists will be well-tolerated. Nonetheless, formal toxicity studies and careful evaluation of human skin treated in clinical trials will be important.

The finding that PAQR7 works through inhibitory $G_i$ subunits is especially interesting, as it is the first example of a melanocyte cellular signaling cascade that *actively represses* melanin synthesis at the level of G-protein signaling, as opposed to classically defined pigment control mechanisms that

modulate the strength of the *stimulatory* MC1R signal. In many animal systems, the Agouti protein decreases pigment production via physically binding to MC1R and inhibiting αMSH stimulation (*Ollmann et al., 1998*), rather than through an actively suppressive mechanism.

Our finding that normal primary melanocytes lack nuclear ER or PR contradicts a prior report (*Im et al., 2002*). This group performed immunohistochemistry and RT-PCR to support the claim that nuclear estrogen receptors are expressed in melanocytes, but in our view, the data presented in that work is not especially convincing, and there is no evidence in that work that nuclear hormone signaling drives changes in melanin synthesis. Another group demonstrated that melanocyte protein extracts have the ability to bind radioactive estrogen, but that work did not identify the specific protein(s) responsible for the binding activity (*Jee et al., 1994*). We do not exclude the possibility that in some tissue settings, including neoplastic lesions and possibly hair follicles, melanocytes express nuclear ER/PR. Still, as there is no known direct signaling pathway linking nuclear sex hormone receptors to the melanin synthesis machinery, it is most likely that the major effects of estrogen and progesterone on pigment production are mediated through the $G_s$ and $G_i$ coupled GPCRs identified in our current work.

We have shown that signaling through GPCRs other than MC1R directly affects melanin production. While surveying all the 7-pass transmembrane proteins expressed in melanocytes, we noted expression of several additional receptors that may also influence melanin production. These include histamine (*Yoshida et al., 2000*; *Lassalle et al., 2003*) and leukotriene receptors, which in other contexts are known to signal through $G_s$ and $G_i$ subunits (*Mondillo et al., 2005*; *Arcemisbéhère et al., 2010*). Future functional analysis of these and other GPCRs in melanocytes may elucidate the mechanisms responsible for the pigmentation changes that frequently accompany many skin diseases associated with inflammation. These studies may identify additional 'drugable' and therapeutically useful receptors in melanocytes, and will help advance an understanding of how cumulative GPCR signaling is integrated to regulate melanin production in human skin.

## Materials and methods

### Melanocyte culture

Primary melanocytes were extracted from fresh discarded human foreskin and surgical specimens as described previously described with some modifications detailed as follows. After overnight incubation in Dispase, the epidermis was separated from the dermis and treated with trypsin for 10 min. Cells were pelleted and plated on selective MC Medium 254 (Invitrogen, Carlsbad, CA) with Human Melanocyte Growth Supplement, and 1% penicillin and streptomycin. Lightly pigmented primary melanocytes were utilized for experiments assaying estrogen and GPER agonist effects, and heavily pigmented primary melanocytes were utilized for experiments assaying progesterone and PAQR7 agonist effects in melanin production. Female iPS-derived human melanocytes were a gift from Meenhard Herlyn (Wistar Institute, Philadelphia, PA, USA). Progesterone (P8783), 17β-Estradiol (E8875), and αMSH (M4135) were purchased from Sigma-Aldrich (St. Louis, MO). G-1 (10008933), G-15 (14673) and G-36 (14397) were purchased from Cayman Chemical (Ann Arbor, MI). CH2P4 (2085) was purchased from Axon Medchem (Groningen, Netherlands). Pertussis toxin was purchased from R&D systems (Minneapolis, MN). These compounds were diluted to working stock solutions in Medium 254.

### Melanin assay

$2 \times 10^5$ melanocytes were seeded uniformly on 6-well tissue culture plates. Cells were treated with vehicle controls, sex steroids, hormone derivatives, or pertussis toxin for 4 days. Cells were then trypsinized, counted, and spun at 300 g for 5 min. The resulting cell pellet was solubilized in 120 μL of 1M NaOH, and boiled for 5 min. The optical density of the resulting solution was read at 450 nm using an EMax microplate reader (Molecular Devices, Sunnyvale, CA). The absorbance was normalized to the number of cells in each sample, and relative amounts of melanin were set based on vehicle treated controls. For tissue melanin assays, tissue was weighed prior to boiling in 1M NaOH for 20 min. Samples were spun down to eliminate insoluble materials, and then the optical density of the sample was measured as previously described and normalized to the weight of tissue.

## Preparation of 3-D organotypic skin cultures

Organotypic skin grafts containing MCs were established using modifications to previously detailed methods (*Ridky et al., 2010*; *Chudnovsky et al., 2005*). The Keratinocyte Growth Media (KGM) used for keratinocyte-only skin grafts was replaced with modified Melanocyte Xenograft Seeding Media (MXSM). MXSM is a 1:1 mixture of KGM, lacking cholera toxin, and Keratinocyte Media 50/50 (Gibco) containing 2% FBS, 1.2 mM calcium chloride, 100 nM Et-3 (endothelin 3), 10 ng/mL rhSCF (recombinant human stem cell factor), and 4.5 ng/mL r-basic FGF (recombinant basic fibroblast growth factor). $1.5 \times 10^5$ melanocytes and $5.0 \times 10^5$ keratinocytes were suspended in 80 μL MXSM, seeded onto the dermis, and incubated at 37°C for 8 days at the air-liquid interface.

## Topical G-1 treatment

2% (w/v) G-1 was prepared in DMSO, 20 μL of this solution was applied daily to the right ear, with vehicle only applied to the left ear, of 4-week-old C57BL/6 mice. These studies were preformed without inclusion/exclusion criteria, randomization, or blinding. Based on a twofold anticipated effect, we preformed this experiment with 3 biological replicates. All procedures were performed in accordance with IACUC-approved protocols at the University of Pennsylvania.

## Western blot analyses and antibodies

Adherent cells were treated with 1 μM doses of E2 and P4 overnight, washed once with DPBS, and lysed with 1% NP-40 buffer (150 mM NaCl, 50 mM Tris, pH 7.5, 1 mM EDTA, and 1% NP-40) containing 1X protease inhibitors (Roche, Basel, Switzerland)) and 1X phosphatase inhibitors (Roche). Lysates were quantified (Bradford assay), normalized, reduced, denatured (95°C) and resolved by SDS gel electrophoresis on 4–15% Tris/Glycine gels (Bio-Rad, Hercules, CA). Resolved protein was transferred to PVDF membranes (Millipore, Billerica, MA) using a Semi-Dry Transfer Cell (Bio-Rad), blocked in 5% BSA in TBS-T and probed with primary antibodies recognizing MITF (Cell Signaling Technology, #12590, 1:1000, Danvers, MA), p-CREB (Cell Signaling Technology, #9198, 1:1000), CREB (Cell Signaling Technology, #9104, 1:1000), and β-Actin (Cell Signaling Technology, #3700, 1:4000). After incubation with the appropriate secondary antibody, proteins were detected using either Luminata Crescendo Western HRP Substrate (Millipore) or ECL Western Blotting Analysis System (GE Healthcare, Bensalem, PA).

## cAMP ELISA

cAMP ELISA was performed on primary human melanocytes using the Cyclic AMP XP Assay Kit (Cell Signaling Technology, #4339) following manufacturer instructions.

## Melanin staining

Formalin-fixed paraffin embedded tissue was sectioned at 5 μM and collected on superfrost plus slides (Fisher, Pittsburgh, PA), and subjected to Fontana-Masson stain for melanin (*Masson, 1928*). Briefly, sections were deparaffinized, rehydrated, and incubated in the following solutions: 2.5% aqueous silver nitrate for 10 min, 0.1% aqueous gold chloride for 15 min, and 5% aqueous sodium thiosulfate for 5 min. Distilled deionized water was used for rinsing and incubations were done at room temperature except for silver nitrate at 60°C. Slides were counterstained with 0.1% nuclear fast red Kernechtrot for 5 min, dehydrated, cleared, and coverslipped using MM24 mounting media (Leica, Wetzlar, Germany). All staining reagents were from Polyscientific R and D Corporation (Bay Shore, NY).

## Immunohistochemistry

Formalin fixed paraffin embedded (FFPE) human skin tissue sections from organotypic tissue was stained for MITF protein expression using a primary antibody to MITF (Leica Biosystems, NCL-L-MITF, 1:15). Staining was performed following the manufacturer protocol for high temperature antigen unmasking technique for immunohistochemical demonstration on paraffin sections.

## Quantification of melanin staining

Tissue sections from organotypic culture were stained using methods described above. Quantification was performed according to *Billings et al. (2015)*. Briefly, 20X photomicrograph images of

representative tissue sections were taken using the Zeiss Axiophot microscope. Tiff files of the images were saved and transferred to Adobe Photoshop where pixels corresponding to Fontana-Masson staining and epidermal counter stain were selected using the color selection tool. Images corresponding to the single specific color were then analyzed using FIJI (Image J) to determine the number of pixels in each sample. The numbers of pixels representing Fontana-Masson staining were normalized to the total amount epidermal counter staining. Final ratios Fontana-Masson staining in the epidermis were set relative to amount of staining in vehicle treated controls.

## Quantitative RT/PCR

mRNA was extracted from melanocytes according to the RNeasy Mini Kit protocol (Qiagen, Venlo, Netherlands), and reverse transcribed to cDNA using the High Capacity RNA-to cDNA kit (Applied Biosystems, Foster City, CA). Quantitative PCR of the resulting cDNA was carried out using Power SYBR Green Master Mix (Applied Biosystems) and gene-specific primers, in triplicate, on a ViiA 7 Real-Time PCR System (Life Technologies). The following primers were used for detection; B-Actin forward: 5'-CAT GTA CGT TGC TAT CCA GGC-3'; B-Actin reverse: 5'-CTCCTTAATGTCACG-CACGAT -3'; ER-A forward: 5'- AAA GGT GGG ATA CGA AAA GAC C -3'; ER-A reverse: 5'-AGC ATC CAA CAA GGC ACT GA-3'; ER-B forward: 5' – GGC TGC GAG AAA TAA CTG CC -3'; ER-B reverse: 5'-AAT GCG GAC ACG TGC TTT TC-3'; PGR forward: 5'- AGG TCT ACC CGC CCT ATC TC -3'; PGR reverse: 5'-AGT AGT TGT GCT GCC CTT CC -3'; AR forward: 5'- GTG CTG TAC AGG AGC CGA AG -3'; AR reverse: 5'- GTC AGT CCT ACC AGG CAC TT -3'; GPER forward: 5'-ACA GAG GGA AAA CGA CAC CT -3'; GPER reverse: 5'- AAT TTT CAC TCG CCG CTT CG -3'; PAQR7 forward: 5'- GTG CAC TTT TAT ACC GTC TGC TT -3'; PAQR7 reverse: 5'- CCT GGG CAG GGA GCT AAG AT -3'. Relative expression was determined using the 2-[delta][delta] Ct method followed by normalization to the AR receptor transcript levels in MCF7 cells.

## Lentiviral vectors

The following shRNAs were expressed from the GIPZ vector and are available through GE Dharmacon (Lafayette, CO). shPAR7.3 (V3LHS_364596, TGTGGTAGAGAAGAGCTGG), shPAQR7.4 (V3LHS_364598, AGAAGTGTGCCAAGGCACT), shGPER.1 (V2LHS_132008, TCCTTC TCCTCTTTAACTC), shGPER.3 (V3LHS_390319, TGATGAAGTACAGGTCGGG). Guide RNAs were designed using software tools developed by the Zhang Lab and provided on the website http://www.genome-engineering.org/ (Hsu et al., 2013). Guide RNAs were subsequently cloned into lenti-CRISPRv2 (Addgene # 52961) according to the accompanying protocol (Sanjana et al., 2014). Guide RNA sequences are as follows: lentiCRISPR GFP 5' GAA GTT CGA GGG CGA CAC CC 3'; lenti-CRISPR GPER.1 5' ACAGGCCGATCACGTACTGC 3'; lentiCRISPR GPER.2 5' GAGCACCAGCAG TACGTGAT 3'; lentiCRISPR PAQR7.1 5' CGTACATCTATGCGGGCTAC 3'; lentiCRISPR PAQR7.5 5' CGTGCGGAAATAGAAGCGCC 3'

## Synthesis of G-1

G-1 (±) 1-(4-(6-bromobenzo[d][1,3]dioxol-5-yl)-3a,4,5,9b-tetrahydro-3H-cyclopenta[c]quinolin-8-yl) ethan- 1-one was prepared by the method of Baudelle et al. (1998) 6-bromopiperonal (1.110 g, 4.85 mmol) and 4-aminoacetophenone (656 mg, 4.85 mmol) were dissolved in anhydrous acetonitrile (16.2 mL, 0.3M) and allowed to stir at 25°C under argon. After approximately 1.5 hr, trifluoroacetic acid (350 µL, 4.61 mmol) was added and the reaction was allowed to stir at 25°C for 45 min. Freshly prepared cyclopentadiene (1.63 mL, 19.4 mmol) was added dropwise to the reaction mixture. After 2 hr at 25°C the reaction mixture was concentrated in vacuo. The crude product was purified by silica gel chromatography using 30% EtOAc in hexanes as eluent to provide racemic 1-(4-(6-bromobenzo [d][1,3]dioxol-5-yl)-3a,4,5,9b-tetrahydro-3H-cyclopenta[c]quinolin-8-yl)ethan-1-one (1.05 g, 53%). The G-1 was >95% pure as determined by high pressure liquid chromatography analysis. 1H and 13C NMR were identical to the data reported by Burai et al. (2010).

## Statistics

*denotes a P-value of less than 0.05 in an unpaired, two-tailed Students T-Test, assuming a normal distribution and equal variance. Due to the anticipated effect size of a twofold change, experiments were performed with 3 biological replicates.

## Acknowledgements

The authors thank Steve Prouty, John Seykora, Christine Marshall, and Aimee Payne from the University of Pennsylvania Skin Disease Research Center for providing primary melanocyte and keratinocyte cultures, and for histologic processing and analysis of tissue sections, William James and Courtney Schreiber for clinical photographs, Jeremy McInerney of the Univ. of Pennsylvania Classics Department for help in interpreting and translating the ancient medical texts, and Zheng Qi for bioinformatics assistance. The authors also thank David Manning, Sarah Millar, George Cotsarelis, John Stanley, Meenhard Herlyn, John Seykora, Thomas Leung, Aimee Payne, and Paul Khavari for critical pre-submission review. TWR is supported by a grant from the NIH/NCI (RO1 CA163566), a Penn/Wistar Institute NIH SPORE (P50CA174523), and the Melanoma Research Alliance. CAN was supported by an NIH/NIAMS training grant (T32 AR0007465-32). EKD received support from an NIH/NIAMS training grant (T32 AR0007465-30) and an NIH/NCI F31 NRSA Individual Fellowship (F31 CA186446).

## Additional information

### Competing interests

CAN is listed as an inventor on patent applications held by the University of Pennsylvania for the use of topical estrogen and progesterone derivatives for modulating skin pigmentation. JDW is listed as an inventor on patent applications held by the University of Pennsylvania for the use of topical estrogen and progesterone derivatives for modulating skin pigmentation. TWR is listed as an inventor on patent applications held by the University of Pennsylvania for the use of topical estrogen and progesterone derivatives for modulating skin pigmentation. The other authors declare that no competing interests exist.

### Funding

| Funder | Grant reference number | Author |
| --- | --- | --- |
| National Cancer Institute | RO1 CA163566 | Todd W Ridky |
| National Cancer Institute | P50CA174523 | Todd W Ridky |
| National Institute of Arthritis and Musculoskeletal and Skin Diseases | T32 AR0007465-32 | Christopher A Natale |
| National Institute of Arthritis and Musculoskeletal and Skin Diseases | T32 AR0007465-30 | Elizabeth K Duperret |
| National Cancer Institute | F31 CA186446 | Elizabeth K Duperret |

The funders had no role in study design, data collection and interpretation, or the decision to submit the work for publication.

### Author contributions

CAN, TWR, Designed experiments, Performed experiments, Analyzed the data, Wrote the manuscript; EKD, JZ, Performed experiments, Analysis and interpretation of data, Drafting or revising the article; RS, AD, Performed experiments, Analysis and interpretation of data; KTO, JDW, Synthesized G-1 for topical treatments, Acquisition of data, Analysis and interpretation of data; RC, Analysis and interpretation of data, Contributed unpublished essential data or reagents

### Author ORCIDs

Christopher A Natale, http://orcid.org/0000-0003-4949-3849
Elizabeth K Duperret, http://orcid.org/0000-0003-1640-4205
Kevin Tyler O'Brien, http://orcid.org/0000-0002-9584-9589
Todd W Ridky, http://orcid.org/0000-0001-8482-1284

### Ethics

Animal experimentation: This study was performed in strict accordance with the recommendations in the Guide for the Care and Use of Laboratory Animals of the National Institutes of Health. All of

the animals were handled according to approved institutional animal care and use committee (IACUC) protocol (#803381) of the University of Pennsylvania.

## Additional files

### Major datasets

The following previously published dataset was used:

| Author(s) | Year | Dataset title | Dataset URL | Database, license, and accessibility information |
|---|---|---|---|---|
| Flockhart RJ, Webster DE, Qu K, Mascarenhas N, Kovalski J, Kretz M, Khavari PA | 2012 | Oncogenic BRAFV600E remodels the melanocyte transcriptome and induces BANCR to regulate melanoma cell migration | http://www.ncbi.nlm.nih.gov/geo/query/acc.cgi?acc=GSE33092 | Publicly available at the NCBI Gene Expression Omnibus (accession no: GSE33092). |

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
