## [Decision Letter]

Thank you for submitting your article "Sex Steroids Regulate Skin Pigmentation through Nonclassical Membrane-Bound Receptors" for consideration by *eLife*. Your article has been reviewed by three peer reviewers, and the evaluation has been overseen by a Reviewing Editor and Fiona Watt as the Senior Editor.

The following individuals involved in review of your submission have agreed to reveal their identity: Jennifer Zhang and Mehdi Rashighi (peer reviewers).

The reviewers have discussed the reviews with one another and the Reviewing Editor has drafted this decision to help you prepare a revised submission.

Summary:

The authors investigated the mechanisms by which sex steroids regulate melanogenesis. The study contains descriptive and mechanistic studies indicating that physiologic estrogen and progesterone exert their differential effects on melanocytes by non-genomic mechanisms. Using genetic and pharmacologic approaches, the authors demonstrated that activation of nonclassical estrogen and progesterone receptors (GPER and PAQR7) are both necessary and sufficient to mediate sex hormone effects on melanocytes.

The effect of sex hormones on skin pigmentation has been observed for many years. There are relatively few studies that investigated the mechanisms that lead to such a phenomenon. A study by McLoad and his colleagues in 1994 reported no staining of ER-α on primary human melanocytes. Interestingly, the observation that estradiol and its 17-α analogue had equivalent effect to increase melanin synthesis, and the surprising similar effect of a pure anti-estrogen compound led them to speculate that the effect of estrogen on melanocytes might be mediated through a "non-classical" mechanism. (Effects of estrogens on human melanocytes in vitro. J Steroid, Biochem. Molec. Biol. 1994).

The current study is well designed, and the experiments follow a logical sequence. The manuscript is also well written.

Essential revisions:

Given that some of the findings contradict previously published observations, the authors need to acknowledge those studies, and briefly discuss them. Notably, some researchers proposed that such inconsistent findings might be due to the fact that the melanocyte response to sex hormones is donor specific:

1) A few studies reported the expression of classical estrogen receptors on melanocytes:

Jee, SH et al. (1994) Effects of estrogen and estrogen receptor in normal human melanocytes. Biochem Biophys Res Commun.;

Tobin, D. J., et al. (2002) Melanocytes in human scalp epidermis and hair follicles express the androgen receptor (AR) and both estrogen receptors (ER-α) and (ER-β). 9th Annual European Hair Research Society Conference (poster presentation);

Im, S., et al. (2002) Donor specific response of estrogen and progesterone on cultured human melanocytes. J. Korean Med. Sci.;

Schmidt AN, et al. (2006). Oestrogen receptor-β expression in melanocytic lesions. Exp Dermatol.

2) A few studies reported the effect of estradiol and progesterone on melanocyte proliferation:

Im, S., et al. (2002) Donor specific response of estrogen and progesterone on cultured human melanocytes. J. Korean Med. Sci.;

Wiedmann, et al. (2008) Inhibitory effects of progestogens on the estrogen stimulation of melanocytes in vitro. Contraception.

There are also a few clarifications that would improve the quality of the manuscript:

1) The authors state that there have been no changes in melanocyte number in the organotypic human tissue. They should also clarify whether treatment of melanocytes in culture had any effect on the proliferation or not.

2) Figure 2—figure supplement 2: The authors should briefly explain why the melanin synthesis is significantly higher in Pertussis toxin alone compared to control.

3) Keratinocytes have been shown to express both classical and non-classical sex steroid receptors (Thornton, M. J. (2005) Oestrogen functions in skin and skin appendages. Expert. Opin. Ther. Targets). Since the authors discuss the therapeutic potential of topical non-classical receptor agonists/antagonist, it would be beneficial to briefly discuss their possible off-target effects. For example, is it likely that GPER agonists might increase the risk of inflammatory skin disorders or skin cancers by transactivation of EGFR on keratinocytes?

---

## [Author Response]

Essential revisions:

Given that some of the findings contradict previously published observations, the authors need to acknowledge those studies, and briefly discuss them. Notably, some researchers proposed that such inconsistent findings might be due to the fact that the melanocyte response to sex hormones is donor specific:

1) A few studies reported the expression of classical estrogen receptors on melanocytes:

*Jee, SH et al. (1994) Effects of estrogen and estrogen receptor in normal human melanocytes. Biochem Biophys Res Commun.; Tobin, D. J., et al. (2002) Melanocytes in human scalp epidermis and hair follicles express the androgen receptor (AR) and both estrogen receptors (ER-α) and (ER-β). 9th Annual European Hair Research Society Conference (poster presentation); Im, S., et al. (2002) Donor specific response of estrogen and progesterone on cultured human melanocytes. J. Korean Med. Sci.; Schmidt AN, et al. (2006). Oestrogen receptor-β expression in melanocytic lesions. Exp Dermatol.* We agree that there is potentially contradicting literature regarding the expression of nuclear estrogen or progesterone receptors in normal melanocytes and melanocytic neoplasms. Jee et al. demonstrated that melanocyte protein extracts have the ability to bind radioactive estrogen, but that work did not identify the specific protein(s) responsible for the binding activity. Im et al. used immunohistochemistry and RT-PCR to support the claim that nuclear estrogen receptors are expressed in melanocytes, but in our view, the data as presented is not convincing, and the techniques used are not quantitative. Critically, those experiments lacked negative controls. We did not examine hair-follicle derived melanocytes, as the cells for our study were derived primarily from foreskins (which are hairless). Although we attempted to review the Tobin et al. poster cited by the reviewer, we were unable to locate any images or corresponding data, and therefore cannot comment directly on the claim in the title of that poster. However, we would certainly be willing to do so if the reviewers were to provide us with the data and methods used to obtain it. To our knowledge there are no peer-reviewed publications providing unequivocal evidence that classical estrogen or progesterone receptors are expressed and functional in normal primary human melanocytes. However, we do not exclude the possibility that in some tissue settings, including neoplastic lesions and possibly hair follicles, melanocytes express nuclear ER/PR. Still, as there is no known direct signaling pathway linking nuclear sex hormone receptors to the melanin synthesis machinery, it is most likely that the major effects of estrogen and progesterone on pigment production are mediated through the G_s_ and G_i_ coupled GPCRs identified in our current work. These points and references are included in the text in the subsection “Estrogen and progesterone reciprocally regulate melanin synthesis” and in the Discussion.

2) A few studies reported the effect of estradiol and progesterone on melanocyte proliferation:

*Im, S., et al. (2002) Donor specific response of estrogen and progesterone on cultured human melanocytes. J. Korean Med. Sci.; Wiedmann, et al. (2008) Inhibitory effects of progestogens on the estrogen stimulation of melanocytes in vitro. Contraception. There are also a few clarifications that would improve the quality of the manuscript: 1) The authors state that there have been no changes in melanocyte number in the organotypic human tissue. They should also clarify whether treatment of melanocytes in culture had any effect on the proliferation or not.* We observed modest changes in melanocyte proliferation when cells were treated with estrogen or progesterone in 2-D monolayer in vitro culture. Estrogen treated melanocytes tended to proliferate slightly slower, while progesterone treated cells tended to proliferate slightly faster. The proliferation effects varied with the basal level of melanin production. Melanocytes from dark skin were more sensitive to progesterone than estrogen, while melanocytes from light skin were more sensitive to estrogen. This is consistent with the idea that more a differentiated melanocyte produces more pigment and proliferates more slowly than a less differentiated melanocyte, but inconsistent with previous studies that have suggested that E2 somehow stimulates proliferation. Nonetheless, these changes in proliferation in monolayer culture are likely an in vitro artifact, as adult epidermal melanocytes are relatively nonproliferative in vivo. In an exhaustive examination of 280 tissue sections of normal human skin from 18 donors, Jimbow et al. identified only 2 proliferative interfollicular melanocytes. Mitotic activity in non-neoplastic melanocytes in vivo as determined by histochemical, autoradiographic, and electron microscope studies. The Journal of Cell Biology. 1975;66(3):663-670. This point, along with our new in vitro proliferation data is now included in new Figure 1—figure supplement 2 and discussed in the subsection “Estrogen and progesterone reciprocally regulate melanin synthesis”.

*2) Figure 2—figure supplement 1: The authors should briefly explain why the melanin synthesis is significantly higher in Pertussis toxin alone compared to control.* Melanin production is dynamic, and likely regulated by many genetic and environmental factors. Melanocytes integrate input from many GPCRs to regulate the level of cAMP/pCREB/MITF signaling that determines the rate of melanin production at any given time. There is likely basal contribution from PAQR7, as well as from other G_i_ coupled GPCRs that contribute to the total cAMP signaling observed in culture. Pertussis toxin treatment inhibits all G_i_ subunits, and this inhibition of the inhibitory inputs is therefore predicted to increase cAMP and downstream pigment production. This point is clarified in the second paragraph of the subsection “Sex steroid signaling in melanocytes is dependent on the nonclassical membrane bound G-protein coupled sex hormone receptors GPER and PAQR7”.

3) Keratinocytes have been shown to express both classical and non-classical sex steroid receptors (Thornton, M. J. (2005) Oestrogen functions in skin and skin appendages. Expert. Opin. Ther. Targets). Since the authors discuss the therapeutic potential of topical non-classical receptor agonists/antagonist, it would be beneficial to briefly discuss their possible off-target effects. For example, is it likely that GPER agonists might increase the risk of inflammatory skin disorders or skin cancers by transactivation of EGFR on keratinocytes?

We recognize the possibility that therapeutic use of GPER or PAQR7 agonists/antagonists could potentially affect cells other than epidermal melanocytes. While topical delivery of such agents would likely avoid off target effects in distant tissues, there exits the theoretical possibility of off-target effects within the skin. However, we did not note any significant abnormalities in the epidermis from our in vitro skin tissues treated with the sex steroids. FDA approved topical estrogen and progesterone products (approved for other non-pigment related indications) have not been associated with increased skin cancer risk or significant skin abnormalities. Interestingly however, topical estrogen has been associated with hyperpigmentation. Given that the increased systemic estrogen and progesterone associated with pregnancy also does not typically result in skin cancer or other non-pigment skin pathology, we think it likely that specific GPER and PAQR7 agonists, will be well-tolerated. Nonetheless, formal toxicity studies and careful evaluation of human skin treated in any future clinical trials will be important. These points are addressed in the Discussion.